# Adsorption of Chromium (III) and Chromium (VI) Ions from Aqueous Solution Using Chitosan–Clay Composite Materials

**DOI:** 10.3390/polym16101399

**Published:** 2024-05-14

**Authors:** Enkhtuya Majigsuren, Ulziidelger Byambasuren, Munkhpurev Bat-Amgalan, Enkhtuul Mendsaikhan, Naoki Kano, Hee Joon Kim, Ganchimeg Yunden

**Affiliations:** 1Department of Chemical Engineering, School of Applied Sciences, Mongolian University of Science and Technology, Ulaanbaatar 14191, Mongolia; menkhtuya@must.edu.mn (E.M.); bulziidelger@must.edu.mn (U.B.); munkhpurev@must.edu.mn (M.B.-A.); f22k503k@mail.cc.niigata-u.ac.jp (E.M.); 2Department of Chemistry and Chemical Engineering, Faculty of Engineering, Niigata University, Niigata 950-2181, Japan; kano@eng.niigata-u.ac.jp; 3Department of Environmental Chemistry and Chemical Engineering, School of Advanced, Engineering, Kogakuin University, Tokyo 192-0015, Japan

**Keywords:** adsorption kinetics, adsorption isotherm, adsorption thermodynamic, chitosan, heavy metal, removal of chromium ion

## Abstract

In this work, biopolymer chitosan and natural clay were used to obtain composite materials. The overall aim of this study was to improve the properties (porosity, thermal stability and density) of pure chitosan beads by the addition of clay and to obtain a chitosan-based composite material for the adsorption of heavy metals from an aqueous solution, using Mongolian resources, and to study the adsorption mechanism. The natural clay was pre-treated with acid and heat to remove the impurities. The chitosan and pre-treated clay were mixed in different ratios (8:1, 8:2 and 8:3) for chemical processing to obtain a composite bead for the adsorption of chromium ions. The adsorption of Cr(III) and Cr(VI) was studied as a function of the solution pH, time, temperature, initial concentration of the chromium solution and mass of the composite bead. It was found that the composite bead obtained from the mixture of chitosan and treated clay with a mass ratio of 8:1 and 8:2 had the highest adsorption capacity (23.5 and 17.31 mg·g^−1^) for Cr(III) and Cr(VI), respectively, in the optimum conditions. The properties of the composite materials, prepared by mixing chitosan and clay with a ratio of 8:1 and 8:2, were investigated using XRD, SEM–EDS, BET and TG analysis. The adsorption mechanism was discussed based on the XPS analysis results. It was confirmed that the chromium ions were adsorbed in their original form, such as Cr(III) and Cr(VI), without undergoing oxidation or reduction reactions. Furthermore, Cr(III) and Cr(VI) were associated with the hydroxyl and amino groups of the composite beads during adsorption. The kinetic, thermodynamic and isothermal analysis of the adsorption process revealed that the interaction between the chitosan/clay composite bead and Cr(III) and Cr(VI) ions can be considered as a second-order endothermic reaction, as such the adsorption can be assessed using the Langmuir isotherm model. It was concluded that the composite bead could be used as an adsorbent for the removal of chromium ions.

## 1. Introduction

Heavy metals pose a high risk to living organisms, due to their toxicity and non-degradability in the aquatic environment [1]. Chromium is a heavy metal that is discharged in wastewater from industries, such as paint and pigment production, stainless steel production, corrosion protection, leather processing, chromium plating and wood preservation. 

There are two stable states of chromium ions in an aqueous solution. One is hexavalent Cr(VI), and the other is trivalent Cr(III). Cr(VI) is 500 times more toxic than Cr(III) [2]. Cr(VI) is persistent in the aquatic environment and can pose significant health risks to humans [1]. The maximum permissible level of total chromium in drinking water is 0.05 ${mg\cdot L}^{-1}$ [3]. 

There are many methods, such as precipitation, ion exchange, adsorption, etc., that can separate heavy and toxic metals, and, among them, the most suitable and selective method for treating samples with low heavy metal content is the adsorption method [4].

Adsorbent materials are classified as either natural or synthetic. However, the resulting synthetic adsorbent material is economically expensive. It is a suitable adsorbent material for low-grade metals due to its high selectivity [5]. Recently, chitosan extracted from the scales of crabs, shrimps and marine mollusks, has been used as an adsorbent for heavy metals [6]. Chitin is the most abundant biopolymer in nature, second only to cellulose. Chitin is N-deacetylated to obtain a bioactive polymer, chitosan (aminopolysaccharide) [7,8]. 

Chitosan is soluble in acidic solutions (less than pH 3), but insoluble in water and alkaline solutions. Most polysaccharides are usually neutral or negatively charged in an acidic environment. Dissolved amino groups ${-NH}_{2}$ of glucosamine are protonated to $-{NH}_{3}^{+}$ [9]. Due to the unique properties of chitosan, such as its electrolytic properties, adhesiveness, hydrophilicity, biodegradability and recyclability, chitosan-based materials are used in the food industry, water purification, bioengineering, cosmetic production and as a support material for chromatography [10,11,12]. 

However, chitosan has disadvantages; it floats due to its low density (close to that of water) and its texture is too soft. These aspects limit its use in practice. New chitosan-based materials are being developed to overcome the disadvantages of pure chitosan [13].

Natural clay is an essential raw material in many applications. Clay of the kaolinite type is a common raw material in soils formed by the chemical weathering of rocks in hot, humid climates and is used in ceramics, cement, medicine, paper, food additives, toothpaste, lamp reflectors, rubber, catalysis and cosmetics. Its large surface area, chemical and mechanical stability, layered structure and high cation exchange capacity make it an excellent adsorbent material [14]. The porosity and adsorption properties of clay can be improved by acid treatment [15].

Several studies have been synthesized and reported on the adsorption parameters of chitosan/clay composites for the removal of heavy metal ions from model water media [16,17,18,19]. Clay is composed of alumina (Al_2_O_3_) and silica (SiO_2_), which act to cross-link chitosan through the silica (SiO_2_) component. It has hydrophilic properties due to its surface hydroxyl -OH groups, which bind silicon, aluminum, calcium and magnesium. The addition of clay to chitosan could improve its density, porosity and thermal stability compared to pure chitosan material. However, more detailed reports are lacking in regard to the need to deeply clarify certain aspects of the adsorption mechanisms of metal ions on chitosan/clay composites. To our knowledge, no literature exists on the investigation of the adsorption mechanism of chromium ions (Cr(III) and Cr(VI)) by chitosan/clay composite beads, both experimentally and theoretically. Therefore, in the present study, the adsorption mechanism was elucidated based on experimental results. 

The overall aim of this study was to improve the properties (porosity, thermal stability and density) of pure chitosan beads by adding a clay and to obtain a chitosan-based composite material for the adsorption of heavy metals from an aqueous solution, using Mongolian resources, and to study the adsorption mechanism.

## 2. Materials and Methods

### 2.1. Materials and Reagents

Chitosan, purchased from Nanjing Dulai Biotechnology Co., Ltd., Nanjing, China, and natural clay, with a particle size of less than 0.04 mm, from the Tsogt-Ovoo deposit in the Umnugovi province of Mongolia, were used as precursors for the synthesis. 

All the chemicals, such as potassium chromate (K_2_CrO_4_) and chromium nitrate (Cr(NO_3_)_3_), ammonium peroxydisulfate (NH_4_)_2_S_2_O_8_, 1,5 diphenyl carbazide (C_13_H_14_N_4_O), sodium hydroxide (NaOH), acetic acid (CH_3_COOH), nitric acid (HNO_3_) and sulfuric acid (H_2_SO_4_), were of analytical grade and were purchased from Xilong Scientific Co., Ltd., Shantou, Guangdong, China. Cr(VI) and Cr(III) standard solutions were prepared from potassium chromate and chromium nitrate. Specifically, 0.01 M NaOH and 0.01 M HNO_3_ were used for pH adjustment. Ultrapure water (>18.2 MΩ in regard to electrical resistance) was treated using the ultrapure water equipment (Barnstead Smart 2 Pure, Thermo Scientific, Atvidaberg, Sweden) used throughout this work.

### 2.2. Apparatus

The mineral and elemental composition of natural clay, acid-treated clay, heat-treated clay, beads prepared from chitosan and a mixture of chitosan and treated clay (composite bead) were analyzed by an X-ray diffractometer (XRD, D2 Phaser, Bruker, Billerica, MA, USA) and an energy dispersive spectrometer (SEM6000-EDS2300, JEOL, Akishima, Tokyo, Japan). Brunauer–Emmett–Teller (BET, Autosorb-IQ, Anton Paar, Ashland, OR, USA) analysis was used to evaluate the surface area, pore volume and pore size of the composite beads. The thermophysical properties of the composite beads were characterized by thermogravimetric differential thermal analysis (ThermoPlus2 TG8120, Rigaku, Tokyo, Japan). 

To confirm the interaction between the chromium ions and the synthesized composite beads, the samples were characterized by X-ray photoelectron spectroscopy (XPS, K-Alpha, Thermo Scientific Center, Waltham, MA, USA) after Cr(III) and Cr(VI) adsorption.

### 2.3. Synthesis of Composite Material

Natural clay, with a particle size of less than 0.04 mm, was treated with sulfuric acid (10%), then the sample was subjected to heat treatment at 1123 K for 1 h to remove the organic mixture [20]. Then, 3 g of treated clay was added to the solution, prepared by dissolving 8 g of chitosan in 200 mL of a 2% acidic acid solution [21] and stirred at a speed of 500 rpm for 6 h to obtain a well-mixed gel solution. The gel was then added, by dropping it into 200 mL of 1 M NaOH solution to obtain the composite bead. 

Due to the reaction of NaOH with the protonated amino groups, the droplets immediately reacted with the chitosan and were stabilized into a spherical shape through agglomeration with the treated clay. This allowed the treated clay gel to coagulate into a spherical shape and become uniform beads [22]. It was then filtered and washed with deionized water until the pH of the filtrate was 7. To be ready for use, the composite bead was dried at room temperature for 12 h and at 333 K for 12 h.

### 2.4. Experiment for the Removal of Chromium

The effect of variables, such as the pH (3–6), time (0.5–6 h), temperature (298–328 K), initial concentration of the chromium solution (10–100 mg·L^−1^) and mass of the composite bead (0.0125–1.0 g), on the adsorption was investigated separately for Cr(III) and Cr(VI). The composite bead was added to a solution containing Cr(III) and Cr(VI), at a known concentration. The mixture was then stirred and filtered. An aliquot of the filtrate was used to determine the Cr(III) and Cr(VI) concentration, using a spectrophotometer (Hitachi U-2910) at a wavelength of 540 nm. The following formula was used to calculate the adsorption capacity.
(1)$$q=\left(C_{0}-C\right)\cdot \frac{V}{m}$$
where *q* is the adsorption capacity, (mg·g^−1^); *C*_0_ and *С* are the initial and equilibrium concentrations of Cr(III) and Cr(VI), (mg·L^−1^); *V* is the volume of the solution, (L); *m* is the mass of the adsorbent, (g). 

The results of the adsorption experiments were used to determine the kinetic, thermodynamic and isothermal parameters of the process.

The adsorption kinetic parameters were calculated using the pseudo-first and pseudo-second-order rate equations [13]. The linearized forms of the pseudo-first-order rate equation and the pseudo-second-order rate equation are given as follows:(2)$$\;\log\left(q_{e}-q_{t}\right)=log\left(q_{e}\right)-K_{1}\frac{t}{2.303}$$
(3)$$\frac{t}{q_{t}}=\frac{1}{K_{2}q_{e}^{2}}+\frac{t}{q_{e}}$$
where *K*_1_ (min^−1^) and *K*_2_ (g·(mg·min)^−1^) are the rate constants of the first and second-order reaction rate constants, $q_{t}$ and $q_{e}$ (mg·g^−1^) are the adsorption capacity at the time *t* and equilibrium, *t* (min) is the adsorption time. The slope and the intercept of the linear plot, for the models, were used to calculate the rate constants (*K*_1_ and *K*_2_) and the adsorption capacity ($q_{e}$, _(cal)_) [23].

The adsorption thermodynamic parameters, such as the changes in Gibb’s standard free energy ∆*G*° (kJ·mol^−1^), the standard enthalpy ∆*H°* (kJ·mol^−1^) and entropy ∆*S*° (J·(mol·K)^−1^), were calculated using the van’t Hoff equation [24].
(4)$$∆G^{0}=-RTlnK_{c}$$
(5)$$lnK_{c}=-\frac{∆H^{0}}{RT}+\frac{∆S^{0}}{R}$$
where *R* is the universal gas constant (8.314 J·(mol·K)^−1^) and *T* is the temperature (K). The value of $K_{c}$ can be obtained by $K_{c}=q_{e}/C_{e}$; $q_{e}$ is the equilibrium adsorption capacity, (${mg\cdot g}^{-1}$); $C_{e}$ is the equilibrium concentration of Cr(III) and Cr(VI), (mg·L^−1^). 

The slope and intercept of the linear relationship between ln$K_{c}$ and 1/*T*, in Equation (5), were used to calculate ∆*H*° and ∆*S*°.

The isothermal parameters were processed using the linearized form of the Langmuir and Freundlich isothermal model equations, as follows:(6)$$\frac{C_{e}}{q_{e}}=\frac{1}{q_{max}b}+\frac{C_{e}}{q_{max}}$$
(7)$$lnq_{e}=lnK_{F}+\frac{1}{n}lnC_{e}$$
(8)$$K_{L}=\frac{1}{1+bC_{0}}$$
where $C_{e}$ is the equilibrium concentration of Cr(III) and Cr(VI), (mg·L^−1^); $q_{e}$ is the equilibrium capacity of the adsorbent, (mg·g^−1^); $q_{max}$ is the maximum adsorption capacity, (mg·g^−1^); *b* is the Langmuir adsorption constant that is related to the energy of sorption, (L·mg^−1^); $K_{L}$(-) is the Langmuir dimensionless constant separation factor, which indicates the nature of the adsorption as either unfavorable ($K_{L}$ > 1), linear ($K_{L}$ = 1), favorable (0 < $K_{L}$< 1) or irreversible ($K_{L}$ = 0). *K*_F_ [(mg·g^−1^)(L·mg^−1^)^1/*n*^ ] and *n* are the Freundlich isotherm constants corresponding to the adsorption capacity and the degree of surface heterogeneity. Moreover, 1/*n* < 1 or 1/*n* ≥ 1 indicates a more heterogeneous or homogeneous surface [25]. The isotherm constants are obtained from the intercept and slope of the linear plots of the isotherm model equations.

### 2.5. Statistical Treatment of the Data

Each adsorption experiment was performed in triplicate and the error bars were calculated to show the deviation between the data points and the mean, as follows:$$\mathrm{Maximum}\;\left\{q_{i}-\bar{q}\right\}\;\;\;\;\;\;\;\;1\leq i\leq 3$$
where $q_{i}$ is the adsorption capacity value of each experiment and $\bar{q}$ is the average adsorption capacity value of the three results. This average is represented by one data point in the figures.

The regression coefficient (*R^2^*) values were considered as a measure of the fit of the experimental data with the models. Moreover, *R*^2^ is a basic statistical criterion that is widely used in the analysis of errors, as follows:(9)$$R^{2}=\frac{\sum y_{i}^{*}y_{i}}{{\left[\sum ({y_{i}^{*})}^{2}\sum ({y_{i})}^{2}\right]}^{0.5}}$$
where $y_{i}$ is the calculated value and $y_{i}^{*}$ is the measured value.

## 3. Results and Discussion

### 3.1. Characterization of the Synthetic Adsorbent

The results of the adsorption experiment showed that the composite beads obtained from a mixture of chitosan and treated clay with a mass ratio of 8:1 and 8:2 had the best adsorption capacity compared to the other mass ratios for Cr(III) and Cr(VI), respectively. Therefore, these composite beads were selected for characterization using instrumental analysis.

#### 3.1.1. XRD Analysis

X-ray diffraction was used to investigate the mineral composition of natural, acid and heat-treated clay (Figure 1a). The major constituents of the untreated clay were dominated by minerals such as quartz SiO_2_, kaolinite Al_2_Si_2_O_5_(OH)_4_, calcite CaCO_3_ and illite (K,H_3_O)Al_2_Si_3_AlO_10_(OH)_2_) . The peaks were in a range of 2θ angle values: 26.68° (110) and 50.10° indicate the presence of illite minerals and quartz; the peaks at 12.59° (020), 20.9° (002) and 25.05° (003) indicate the presence of kaolinite minerals. Also, the peaks at 28.01°, 29.63° (113) and 39.51° were for calcite, and the peaks at 8.7° (004) and 26.6° were for illite. After acid and heat treatment, the main peaks shifted to the low-angle side, which indicates a change in the lattice structure of the clay crystals and an increase in the interatomic space in the crystals. Also, new peaks (14.72°, 31.93°) appeared on the XRD graph and some mineral peaks disappeared. This confirmed that kaolinite was extracted from the natural clay [26]. 

Figure 1b shows the XRD analysis results of the beads obtained from pure chitosan and a mixture of chitosan and treated clay. According to the results of the XRD analysis, for the pure chitosan bead, the main broad peak occurred in a range of 2θ angle values, 10.00° (020) and 20.00° (200), which indicates the presence of chitosan. However, for the beads obtained from a mixture of chitosan and treated clay with a ratio of 8:2 and 8:1, the peaks of chitosan occurred at 2θ angles values, 10.00° (020) and 20.00° (200); the peaks of illite at 26.6° (020); and the peaks of quartz at 50.01° (003). These results confirm that the beads were composed of a mixture of chitosan and treated clay [27].

#### 3.1.2. SEM/EDS Analysis

The surface morphology and elemental composition of natural clay, acid and heat-treated clay were characterized by SEM/EDS analysis. According to the SEM analysis, no significant changes were observed in the morphology. According to the results of the EDS analysis (Table 1), after the acid treatment, the Mg content was not detected, and the Ca content was decreased due to minerals containing Mg and Ca dissolved in sulfuric acid, and the content of Al and Si was increased. 

However, when subjected to heat treatment, after the acid treatment, the Al and Si content in the clay sample decreased compared to the clay sample after the acid treatment, while the Ca content increased. This confirms that the Ca crystals containing clay minerals were reshaped and exposed on the surface of the clay during heat treatment. The SEM images of the pure chitosan bead and composite beads (prepared from 8:1 and 8:2 ratio mixtures of chitosan and treated clay) are compared in Figure 2. 

It can be seen that the surface of the pure chitosan bead, with an average diameter of 1.1–1.2 mm, has a uniform texture and is smooth, whereas the surfaces of the composite beads were not smooth due to structural changes. The SEM images of the composite beads loaded with Cr(III) and Cr(VI) after adsorption are compared in Figure 3. 

Chitosan is soluble in acidic conditions. Therefore, in order to observe the solubility of chitosan during adsorption under the conditions in this experiment, the size changes before and after adsorption were observed in the SEM photographs. The SEM images of the composites before and after adsorption showed that the shape of the adsorbed material did not change, and the size did not shrink. This means that in our experimental conditions, the chitosan did not dissolve during the adsorption process to the extent that it affected the experiment. 

#### 3.1.3. BET Analysis

Brunauer–Emmett–Teller analysis was used to evaluate the surface area, pore volume and pore size of the materials. The results showed that the treated clay reduced in terms of its surface area, while the pore volume and pore size increased (Table 2). This confirms that the treated clay, not the cross-linking, affects the pore size and volume. Although the specific surface area was decreased, this should not have a significant effect on the adsorption capacity, since the pore size and volume were increased in the adsorption experiments. On the other hand, an increase in porosity could actually have a positive effect on the permeation flux of water [28].

#### 3.1.4. TGA Analysis

The composite materials were characterized using thermogravimetric (TG) analysis. The TG and DTA curves are shown in Figure 4. 

We found that the total mass loss of the new material prepared from a mixture of chitosan and treated clay with mass ratios of 8:0, 8:1 and 8:2 was 98.2%, 90.9% and 78%, respectively, which means that it is directly related to the mass ratio of treated clay in the adsorbent material. In other words, the composite adsorbent obtained from a mixture of chitosan and treated clay with mass ratios of 8:1 and 8:2 contains approximately 10% and 20% of treated clay in the material. The temperature range of the main combustion was between 150–500 ℃ [29]. From Figure 4, we can see that the decomposition temperature is related to the content of the treated clay, between 573 K and 773 K in the composite material. These results indicate that the thermal stability and density are improved by mixing chitosan with treated clay, due to the strong electrostatic attraction between the chitosan and the cationic metals in the treated clay.

### 3.2. Adsorption Mechanism

The XPS analysis results of the composite beads after Cr(III) and Cr(VI) adsorption, prepared from a mixture of chitosan and treated clay with mass ratios of 8:1 and 8:2 are shown in Table 3, and Figure 5 and Figure 6. The results of the XPS analysis confirmed that the mixing ratio of chitosan and treated clay affects the content changes. For example, the C and N content is higher in the 8:1 composite bead, and the O and Si content is higher in the 8:2 composite bead. In other words, only chitosan contains C and N. As the ratio of chitosan in the bead absorbent increased, the C and N content increased. In addition, as the amount of treated clay increased during the preparation of the composite bead, the O and Si content increased, as shown in Table 3.

In terms of C, C–C and C–NH_2_ bonds are dominant in the 8:1 composite bead [29]. It can be concluded that this bond in chitosan remained unbroken when treated clay was added, in a small amount, during the preparation of the composite bead. On the other hand, C–O, C–O–C, C=O and O–C=O bonds are dominant in the 8:2 composite bead, which means that as the amount of treated clay in the adsorbent increases, the C atoms of chitosan are likely to be bonded to the O atoms of the treated clay. The C–NHR bond was also detected in the 8:2 composite bead. As for the O in the absorbent, -OH and -COOH are the dominant groups in the 8:1 and 8:2 composite beads, respectively. When the treated clay ratio is low, the hydroxyl group remains in the chitosan; however, when the treated clay ratio is higher, the hydroxyl group in the treated clay is affected by oxygen and becomes the carboxyl group [30]. Also, it can be confirmed that the -OH group may have been added to the water molecules in the treated clay in regard to the 8:2 composite bead. In regard to N in the adsorbent, the C–N bonds in the 8:1 composite bead are greater than those of the 8:2 composite bead, which is directly related to the ratio of chitosan to treated clay. 

However, the high content of ${NH}_{3}^{+}$ ions in the 8:2 composite bead is attributed to the fact that ${NH}_{2}$, or the amino group in chitosan, is protonated and converted into ${NH}_{3}^{+}$ in the acidic environment during adsorption [31]. In regard to Si in the adsorbent, Si–O–C bonds are dominant, suggesting that carbon atoms are connected to silicon through oxygen bridges during the interaction between the chitosan and treated clay. The Si–O–C bonds in the 8:1 composite bead are greater in number than those of the 8:2 composite bead, which indicates that when the amount of chitosan in the adsorbent is high, there is a sufficient amount of carbon atoms to bond with the silicon in the treated clay [32]. However, the high SiO_2_ content in the 8:2 composite bead indicates that as the amount of treated clay in the adsorbent increases, more silicon atoms have not yet participated in the bonding. The XPS analysis shows that the chromium ions are adsorbed in their original form, Cr(III) and Cr(VI), respectively, without undergoing oxidation or reduction reactions. When Cr(III) and Cr(VI) were adsorbed on the composite bead, Cr(III) could be associated with the hydroxyl group of the adsorbent, while Cr(VI) was associated with an amino group of the composite bead [4].

### 3.3. Adsorption Properties 

#### 3.3.1. Effect of pH

We conducted the adsorption experiment in the pH range of 3–6, because the chitosan bead decomposes when the pH is less than 2. In an aqueous solution, the dominant ions for Cr(III) and Cr(VI) are Cr^3+^ and НCrO_4_^−^, respectively [33]. The adsorption capacity of the composite beads prepared from a mixture of chitosan and treated clay with a mass ratio of 8:1, 8:2 and 8:3 were compared with the bead prepared from pure chitosan. Figure 7a,b shows the dependence of the pH on the Cr(III) and Cr(VI) adsorption amount by the synthesized adsorbents and the dependence of the mass ratio of chitosan/clay for the same pH condition. In the pH range of 3–6, the dependence of the pH on the Cr(III) adsorption amount by the adsorbent is smaller than that of Cr(VI).

Under the same mass ratio of chitosan/clay, the difference in the Cr(III) adsorption amount is less than 5%, even if the pH changes from 3 to 6. However, the adsorption amount of Cr(VI) is highly pH dependent, and the Cr(VI) adsorption amount at pH 6 is less than 1/3 of that at pH 3. For the mass ratio of chitosan/clay, the adsorption amount of Cr(III) decreases with increasing clay content at pH 3 and pH 4 and becomes almost the same at above pH 5. However, the Cr(VI) adsorption amount tends to decrease with increasing clay content in the pH range of 3–6. This could be related to the deprotonation of the adsorbent surface as the pH is increased. In other words, we can predict that the dominant functional groups for the attraction between the adsorbent and the chromium ions are -OH and ${-NH}_{2}$ for Cr(III) and Cr(VI), respectively. Another aspect is predicted to be due to the difference in adsorption capacity, whether + trivalent for Cr(III) and − univalent for Cr(VI).

For the pH condition in the next adsorption experiments, pH 4 and 3 were chosen for Cr(III) and Cr(VI), respectively, and the mass ratio of chitosan and treated clay was 8:1 and 8:2 for Cr(III) and Cr(VI), respectively.

#### 3.3.2. Effect of Competitive Cations 

The experiment to study the effect of competitive metals was carried out with an optimized pH using different concentrations of Na^+^, K^+^, Ca^2+^ or Mg^2+^, and a combination of all metal ions (i.e., 0, 10, 25 and 50 mg$\cdot L^{-1}$). It can be seen that the adsorption capacity decreased as the competitive metal ion concentration increased (Figure 8). However, no significant decrease in the adsorption capacity was observed.

#### 3.3.3. Adsorption Kinetics

The experiments for the kinetic study were carried out at 298 K. Based on the results of the previous experiment, the pH of the solutions was adjusted to 4 and 3 for the Cr(III) and Cr(VI) solutions, respectively (Figure 9). 

The experimental results were processed by the first and second-order models [34,35], and the results are compared and summarized in Table 4. From the results, it can be seen that the adsorption capacity of the composite bead increased with the increase in the adsorption time, and the equilibrium was established around 240 and 180 min for Cr(III) and Cr(VI), respectively. Therefore, 240 min for Cr(III) and 180 min for Cr(VI) were considered suitable parameters of the adsorption kinetics. As seen in Table 4, for the adsorption of Cr(III) and Cr(VI) in the above cases, the values of the correlation coefficient of the second-order model are higher than the correlation coefficient (*R*^2^) of the first-order model. The value of the adsorption capacity calculated by the second-order model is relatively close to the experimental results. Therefore, the interaction between the composite bead and Cr(III) and Cr(VI) ions may be a second-order reaction [36,37].

#### 3.3.4. Adsorption Thermodynamics

Usually, in adsorption thermodynamic studies, the temperature value is chosen between 328 K and 338 K. In our case, the adsorption experiment was carried out in the temperature range of 298–328 K. Figure 10 shows that the adsorption capacity increases slightly as the temperature increases. This can be explained as an increase in the adsorption capacity due to an increase in the adsorbate (Cr(III) or Cr(VI)) migration rate and, consequently, an increase in the number of collisions effective for adsorption as the temperature increases [38]. 

Figure 10 shows that the adsorption capacity does not increase significantly with an increase in the temperature. From this result and from an economic point of view, room temperature is considered suitable for adsorption isothermal experiments. Therefore, the following isothermal research experiments were conducted at 298 K. 

The results were used for the determination of the thermodynamic parameters, such as the Gibbs free energy change (ΔG°), enthalpy change (ΔH°) and entropy change (ΔS°) [39], and the results are summarized in Table 5. 

Table 5 shows that the negative value of the thermodynamic parameter ΔG° tends to increase when the temperature of the adsorption processes increases, which indicates that the process is suitable for low temperatures and that the process is spontaneous. A positive value of ΔH° indicates that an endothermic process occurred in the system [40]. The positive value of ΔS° proves the disorderliness of the adsorption on the solid–liquid interface [41].

#### 3.3.5. Adsorption Isotherm 

The experiment on the effect of the adsorbent mass on the adsorption process was carried out at 298 K, when the initial concentration was 50 mg·L^−1^. The adsorbent mass was in the range of 0.0125–1.0 g (Figure 11). It was found that as the adsorbent dosage increased, the removal efficiency increased as more adsorption sites were provided. However, the adsorption capacity decreased as the adsorbent dosage increased. The removal efficiency increased from 32.7% to 90.0% and from 31.5% to 96.2% for Cr(III) and Cr(VI), by increasing the composite bead dose from 0.0125 to 1.0 g. 

The experimental data were fitted to the Langmuir and Freundlich isotherm linear plots, 1/*q_e_* versus 1/*C_e_* and ln*q_e_* versus ln*C_e_* [42]. The *R*^2^ values were considered as a measure of the fit of the experimental data with the isothermal models. The parameters of the Langmuir and Freundlich isotherm models for the metal ions are summarized in Table 6. As observed in Table 6, the process of Cr(III) and Cr(VI) adsorption onto the composite bead is more consistent with the linear fitting of the Langmuir isotherm model because its regression coefficients (*R^2^* = 0.998 and 0.977) are higher than those of the Freundlich isotherm model (*R^2^* = 0.981 and 0.899). In other words, it was concluded that monolayer adsorption was the dominant adsorption mode in this system.

The comparison results of the adsorption capacity of chitosan/clay composites with other reported adsorbents are illustrated in Table 7. As seen in Table 7, the adsorption capacity of the chitosan/clay composite in the present study for Cr(VI) was better than that of some adsorbents in previous studies, but lower for Cr(III). A feature of this study is the simple and low-cost method for the production of adsorbents for chromium (III) and chromium (VI).

## 4. Conclusions

In this work, a composite bead for the adsorption of chromium ions was obtained using pure chitosan and natural clay from the Tsogt-Ovoo deposit in Mongolia. The XRD, SEM/EDS, BET and TG analysis results confirmed that the composite material was successfully synthesized. The results of the experiment to study the effect of pH confirm that the adsorption capacity of the composite beads prepared from a mixture of chitosan and treated clay with a ratio of 8:1 and 8:2 was the highest at pH = 4 and pH = 3 for Cr(III) and Cr(VI) adsorption, respectively. The kinetics, thermodynamics and isothermal analysis of the adsorption process revealed that the interaction between the composite bead and Cr(III) and Cr(VI) ions can be considered as a second-order endothermic reaction, and the adsorption can be assessed using the Langmuir isotherm model. The XPS results confirm that the chromium ions were adsorbed in their original form, Cr(III) and Cr(VI), respectively, without undergoing oxidation or reduction reactions. Cr(III) could be associated with the hydroxyl group of the composite bead. Cr(VI) is associated with the amino group of the composite bead during adsorption.

## Figures and Tables

**Figure 1 polymers-16-01399-f001:**
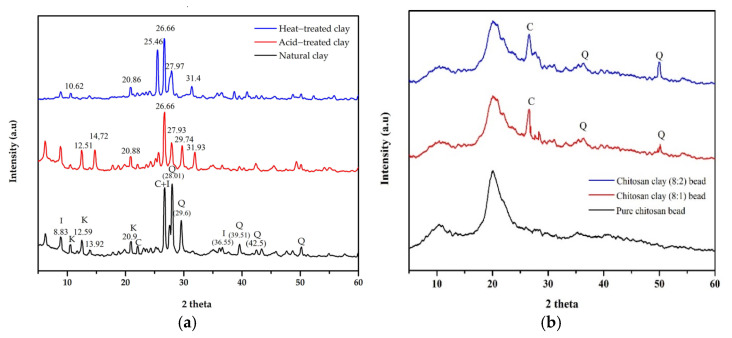
XRD pattern: (**a**) natural and treated clay; (**b**) pure chitosan bead and composite bead [I—illite, K—kaolinite, C—calcite and Q—quartz].

**Figure 2 polymers-16-01399-f002:**
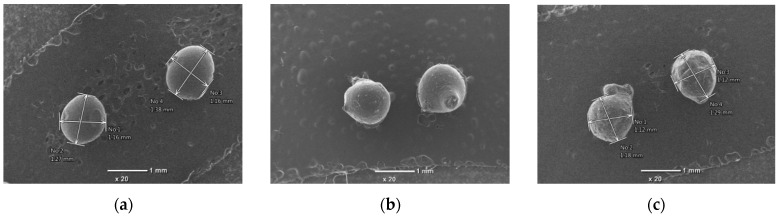
SEM images of the beads before adsorption: (**a**) pure chitosan bead, (**b**) chitosan-treated clay bead (8:1), (**c**) chitosan-treated clay (8:2) bead. Transmittance ×20.

**Figure 3 polymers-16-01399-f003:**
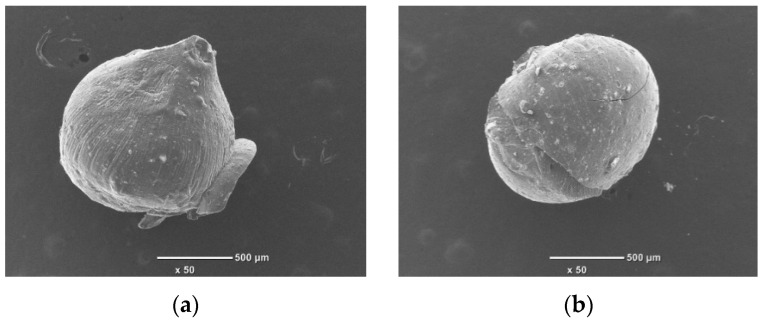
SEM images of composite beads after adsorption: (**a**) composite bead (8:1) loaded with Cr(III) and (**b**) composite bead (8:2) loaded with Cr(VI). Transmittance ×50.

**Figure 4 polymers-16-01399-f004:**
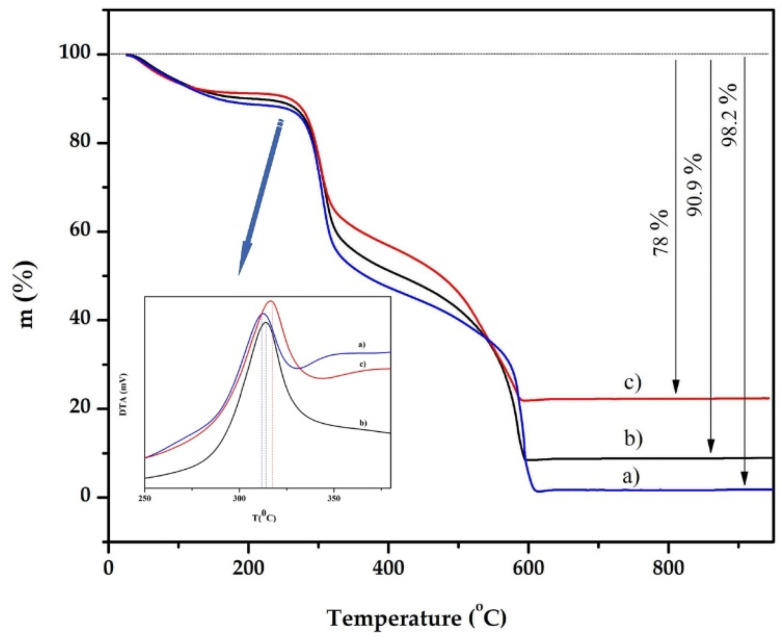
TGA curves of: (a) pure chitosan bead, (b) chitosan–clay (8:1) bead and (c) chitosan–clay (8:2) bead.

**Figure 5 polymers-16-01399-f005:**
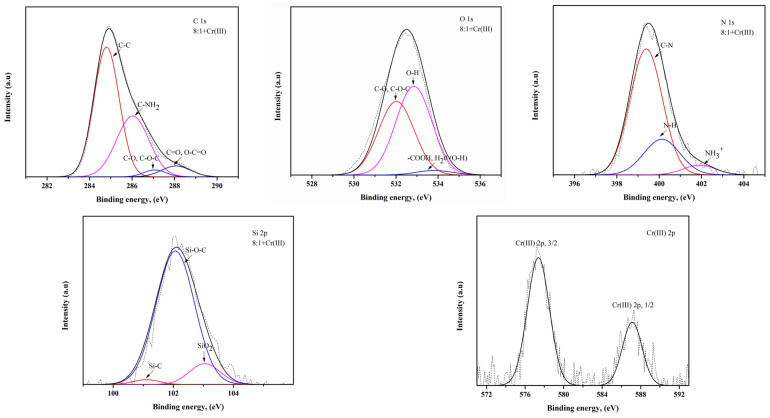
XPS spectra of C 1s, N 1s, O 1s, Si 2p and Cr(III) 2p for the bead prepared from a mixture of chitosan and treated clay with a mass ratio of 8:1 after the adsorption of Cr(III).

**Figure 6 polymers-16-01399-f006:**
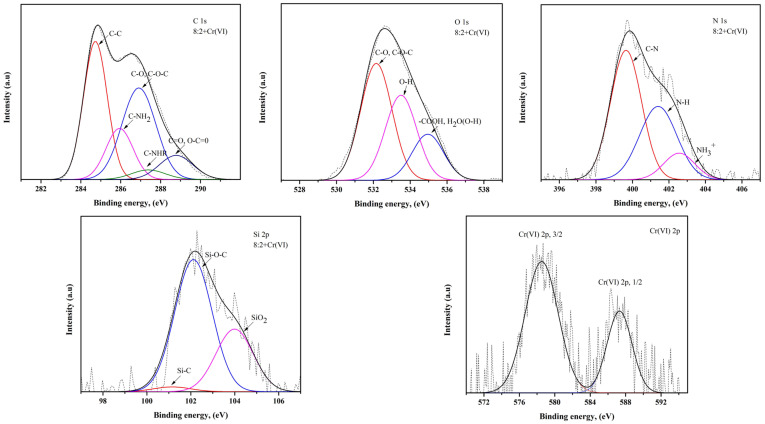
XPS spectra of C 1s, N 1s, O 1s, Si 2p and Cr(VI) 2p for bead prepared from a mixture of chitosan and treated clay with a mass ratio of 8:2 after the adsorption of Cr(VI).

**Figure 7 polymers-16-01399-f007:**
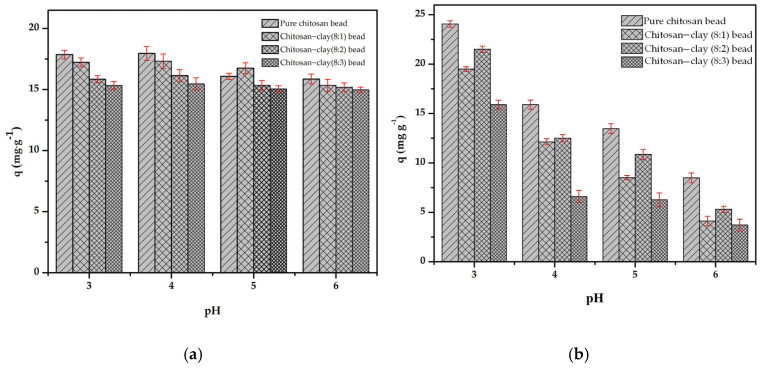
Effect of pH on the adsorption of: (**a**) Cr(III) and (**b**) Cr(VI) onto chitosan bead and chitosan–clay composite bead (t = 120 min, T = 298 K, C_0_ = 50 mg·$L^{-1}$, V = 50 mL, m = 0.1 g, n = 100 rpm).

**Figure 8 polymers-16-01399-f008:**
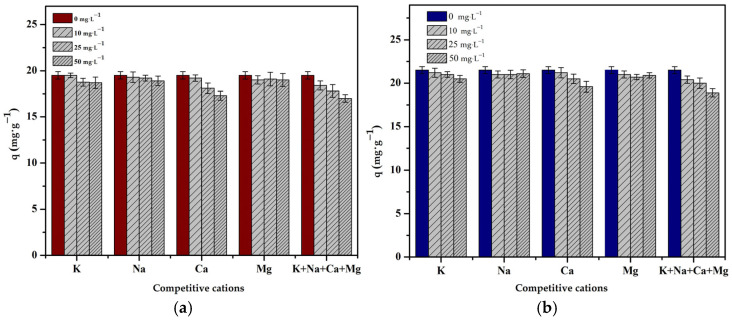
Effect of competitive cations on the adsorption capacity: (**a**) Cr(III) and (**b**) Cr(VI) onto chitosan bead and chitosan–clay composite bead (pH = 4 and 3, t = 240 and 180 min, T = 298 K, C_0_ = 50 mg·L^−1^, V = 50 mL, m = 0.1 g, n = 100 rpm).

**Figure 9 polymers-16-01399-f009:**
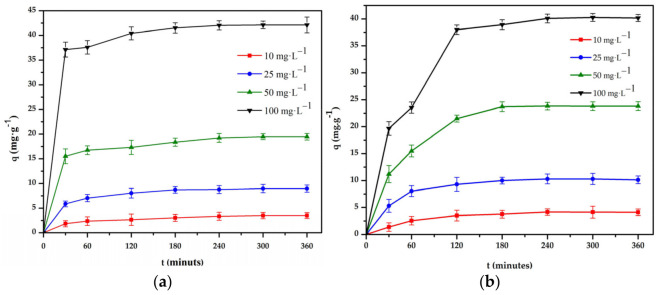
Effect of time on the adsorption of: (**a**) Cr(III) and (**b**) Cr(VI) onto chitosan bead and chitosan–clay composite bead (T = 298 K, pH = 4 and 3, V = 50 mL, m = 0.1 g, n = 100 rpm).

**Figure 10 polymers-16-01399-f010:**
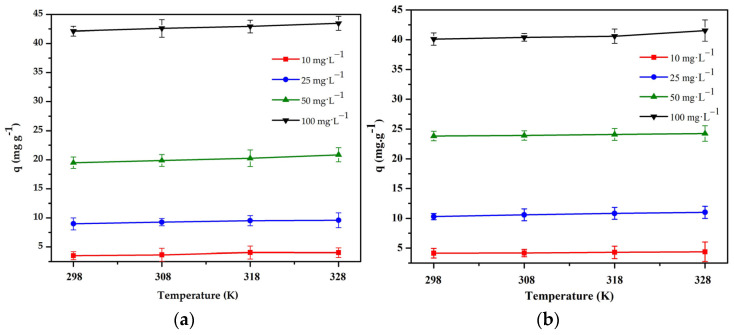
Effect of temperature on the adsorption of: (**a**) Cr(III) onto chitosan and clay (8:1) bead and (**b**) Cr(VI) onto chitosan and clay (8:2) bead (pH = 4 and 3, t = 240 and 180 min, V = 50 mL, m = 0.1 g, n = 100 rpm).

**Figure 11 polymers-16-01399-f011:**
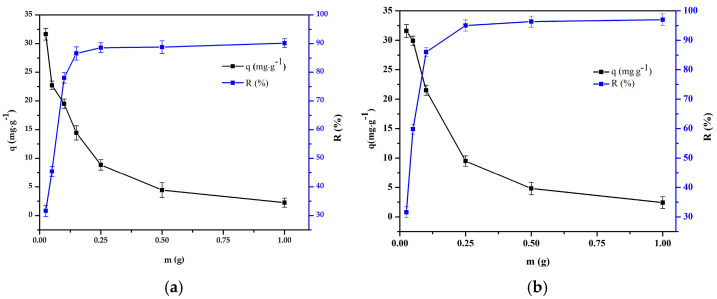
Effect of adsorbent mass on the adsorption of: (**a**) Cr(III) onto chitosan and clay (8:1) bead and (**b**) Cr(VI) onto chitosan and clay (8:2) bead (pH = 4 and 3, t = 240 and 180 min, T = 298 K, C_0_ = 50 mg·L^−1^, V = 50 mL, n = 100 rpm).

**Table 1 polymers-16-01399-t001:** Elemental content.

Sample	Mass, %			
	Mg	Al	Si	Ca
Natural clay	14.90	17.56	44.89	25.56
Acid-treated clay	-	20.13	60.64	19.22
Heat-treated clay	-	15.82	58.32	25.85

**Table 2 polymers-16-01399-t002:** Surface properties of pure chitosan and chitosan–clay composite bead.

Sample	Surface Area (m^2^·g^−1^)	Pore Volume (cc·g^−1^)	Pore Size (nm)
Pure chitosan bead	5.802	0.047	16.37
Chitosan–clay (8:1) bead	2.157	0.069	64.66
Chitosan–clay (8:2) bead	2.525	0.136	108.3

**Table 3 polymers-16-01399-t003:** XPS analysis results after adsorption.

Elements	8:1 Chitosan–Clay Bead (Cr(III))			8:2 Chitosan–Clay Bead (Cr(VI))		
	Atomic, %	Sub-Peak	Ratio, %	Atomic, %	Sub-Peak	Ratio, %
C	71.88	C-C	57.34	71.23	C-C	36.77
		C-NH_2_	34.56		C-NH_2_	15.56
		C-O, C-O-C	2.61		C-O, C-O-C	34.83
		C-NHR	-		C-NHR	3.64
		C=O, O-C=O	5.46		C=O, O-C=O	9.17
O	19.06	C-O, C-O-C	53.18	20.36	C-O, C-O-C	47.11
		OH	44.10		OH	34.30
		COOH, H_2_O(O-H)	2.70		COOH, H_2_O(O-H)	18.58
N	4.98	C-N	71.91	4.39	C-N	53.04
		N-H	22.58		N-H	36.05
		NH_3_^+^	5.49		NH_3_^+^	10.89
Cr	1.61	Cr(III) 2p3/2	69.48	0.94	Cr(VI) 2p3/2	68.14
		Cr(III) 2p1/2	30.51		Cr(VI) 2p1/2	31.85
Si	2.48	Si-C	2.06	3.08	Si-C	2.41
		Si-O-C	87.09		Si-O-C	66.19
		SiO_2_	10.83		SiO_2_	31.39

**Table 4 polymers-16-01399-t004:** Kinetic parameters of Cr(III) and Cr(VI) adsorption.

Ions	*C*_0_, mg·L^−1^	*q*_e_ (exp), mg·g^−1^	Pseudo First Order			Pseudo Second Order		
			*K*_1_, min^−1^	*q*_e_ (cal), mg·g^−1^	*R^2^*	*K*_2_, g·(mg·g)^−1^	*q*_e_ (cal), mg·g^−1^	*R^2^*
Cr(III)	10	3.51	0.106	4.74	0.862	0.347	3.85	0.991
Cr(VI)		4.16	0.113	5.32	0.871	1.192	5.03	0.991
Cr(III)	25	8.98	0.111	2.82	0.763	0.332	9.09	0.999
Cr(VI)		10.31	0.124	8.65	0.892	0.165	12.50	0.998
Cr(III)	50	19.49	0.114	13.82	0.873	0.216	20.20	0.998
Cr(VI)		23.82	0.150	26.07	0.901	0.061	27.03	0.994
Cr(III)	100	42.12	0.116	11.53	0.839	0.208	43.48	0.999
Cr(VI)		40.11	0.156	53.95	0.881	0.033	45.46	0.991

**Table 5 polymers-16-01399-t005:** Thermodynamic parameters of Cr(III) and Cr(VI) adsorption.

T, K	*C*_0_, mg·L^−1^	ΔG° , ${kJ\;mol}^{-1}$		ΔH° , ${kJ\;mol}^{-1}$		ΔS° , $J\;{mol}^{-1}K^{-1}$	
		Cr(III)	Cr(VI)	Cr(III)	Cr(VI)	Cr(III)	Cr(VI)
298	50	−1.418	−5.728	9.33	14.68	35.91	58.43
308		−1.699	−6.258				
318		−2.012	−6.999				
328		−2.522	−7.607				

**Table 6 polymers-16-01399-t006:** Parameters of Langmuir and Freundlich isotherm models.

Ions	Langmuir			Freundlich		
	$q_{\max},\;$ ${mg\cdot g}^{-1}$	*K* _L_	*R^2^*	$K_{F},$ ${mg\cdot g}^{-1}$(L·mg^−1^) ^1/*n*^	1/*n*	*R^2^*
Cr(III)	26.95	0.140	0.998	15.131	0.650	0.981
Cr(VI)	90.91	0.035	0.977	20.893	0.730	0.899

**Table 7 polymers-16-01399-t007:** Comparison of adsorption capacity of chitosan-treated clay beads with other reported adsorbents.

Adsorbents	pH	Ion	*q*_max_ mg·g^–1^	Ref
Chitosan beads modified with sodium dodecyl sulfate (SDS)	4	Cr(VI)	3.2	[6]
Clay	7	Cr(VI)	13.9	[43]
Surfactant-modified bentonite	3.4	Cr(VI)	10.0	[44]
Magnetic kaolin-embedded chitosan (MKCS) beads	3.02	Cr(VI)	144.0	[4]
Acid-activated kaolin AAK	4	Cr(VI)	50.2	[15]
SDS–chitosan	4	Cr(III)	3.4	[34]
Graphene oxide/alginate hydrogel membrane GAHMs	6	Cr(III)	118.6	[45]
Fe_3_O_4_ NPs decorated with MoS_2_(MoS_2_@Fe_3_O_4_NPs)	5	Cr(VI)	218.2	[45]
Chitosan	3.8	Cr(III)	138.0	[41]
Amine-functionalized zeolite	3	Cr(VI)	13.5	[46]
Aminated cross-linked chitosan beads (CS–DEO–SP)	2	Cr(VI)	358.2	[47]
Chitosan–clay biocomposite beads	3	Cr(VI),	50.9	[16]
Chitosan–clay composite bead	4 3	Cr(III) Cr(VI)	26.9 90.9	In this study

## Data Availability

The data presented in this study are available on request from the corresponding author. The data are not publicly available due to privacy and ethical reasons.

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
