# Peer review of "Adsorption of Chromium (III) and Chromium (VI) Ions from Aqueous Solution Using Chitosan–Clay Composite Materials"

_polymers, 2024, doi:10.3390/polym16101399_

Round 1

Reviewer 1 Report

Comments and Suggestions for Authors

Comments regarding the manuscript entitled "Research on the synthesis of a composite material for chromium adsorption from aqueous solution" (Ref.: polymers-2972553).

Considering the interest in the possible utilization of novel adsorbents for the bioremediation of Cr(III)- and Cr(VI)-polluted waters and wastewaters, new information on this topic is welcome. In this context, the paper's subject is interesting because of its implications for health and the environment. However, some criticisms can be made of the present version of the manuscript. My principal comments are as follows:

1)    The manuscript needs revision regarding the English language to improve the quality of communication. 

2)    Abstract: Provide information on kinetic, equilibrium, and thermodynamic studies of Cr(III) and Cr(VI) adsorption results.

3)    In their response and revised manuscript, the authors must clarify their work's novelty compared to the state of the art in the literature on other adsorption systems. Several papers are found in the literature on the synthesis of chitosan-clay composites for the adsorption of heavy metals.

4)    The Materials and Methods section needs more details to allow others to evaluate and reproduce the experiments. For example, no information is provided about the conditions used for each adsorption experiment (pH, temperature, adsorbent dosage, initial metal concentration, agitation speed, shaking time, etc.), preparation of the chitosan-clay composites, etc.

5)    Statistical treatment of data needs to be included.

6)    Figure 1: Please define Q, C, etc.

7)    Line 208: Please revise the following statement: the dominant ions for Cr(III) and Cr(VI) are HCrO4- and Cr3+ respectively.

8)    There appears to be no statistically significant difference in some of the results (for example, Figure 5a and 6). Analyze and discuss your results based on the statistically significant difference.

9)    Figure 5: The Cr(VI) and Cr(III) adsorption capacities of pure chitosan beads are higher than those of the chitosan-clay composites. What are the advantages of using chitosan-clay composites for chromium adsorption purposes? 

10) In addition to a lower adsorption capacity of Cr(VI) and Cr(III), what are the disadvantages of chitosan-clay composites?

11) Section 3.2.2: Please show the kinetic profiles of variation of the adsorption capacity of Cr(VI) and Cr(III) as a function of time at the different initial concentrations of Cr(III) and Cr(VI) tested for a better understanding of the process.

12) Why do the rate constant K2 values increase and decrease as the initial concentration of Cr(III) increases? This trend is rare. Provide possible explanations for this behavior.

13) There seem to be no significant statistical differences in the equilibrium adsorption capacity at the different temperatures tested, which is very common. However, temperature affects the adsorption rate. How did temperature affect Cr(III) and Cr(VI) adsorption rates?

14) Please provide the activation energy for the adsorption of Cr(III) and Cr(VI) onto the chitosan-clay composites tested?

15) Figure 7: Define R(%).

16) Equations 6 and 7: Define qmax and Ce.

17) Line 286 and Table 5: No units are provided for KL and KF.

18) Please provide the isotherms for Cr(III) and Cr(VI) so readers can better understand the equilibrium processes. What type of isotherms they are?

19) Lines 297-298: What term do the authors refer to when they say it represents the degree of heterogeneity and the energy distribution of adsorption sites? It needs to be clarified.

Comments on the Quality of English Language

The manuscript needs revision regarding the English language to improve the quality of communication.

Author Response

Thank you for your excellent review. Due to the large number of answers, I will send them as attachments.

Reviewer 2 Report

Comments and Suggestions for Authors

Review report

Dear authors, I have read your review paper entitled: Research on The Synthesis of a Composite Material for Chromium Adsorption from Aqueous Solution, my suggestions are:

1.         Rename the title so that it attracts the reader's attention.

2.         In the abstract, it is necessary to write the main objective representation of the paper.

3.         An introduction showing the novelty of composite material was expected and essential to development.

4.         The materials and methods section must be completed. The methodology is presented in a disorganized, erroneous, and imprecise manner. There is a lack of information on the reagents and materials used, the characteristics of the solutions to be used and their concentrations, the techniques, the procedures, etc.

5.         Add equations 2-7 to materials and methods to have uniformity in the presentation of the article.

6.         You started with three formulations (pre-treated chitosan and clay were mixed with various ratios (8:1, 8:2, and 8:3) for chemical processing to obtain a composite bead for the adsorption of chromium ions, and in the majority of cases are presented only formulations of 8:1, 8:2 at sections 3.1 for characterization of synthetic adsorbent. It must be explained in the article why you present 3 and characterize only 2. At the same time, and in the abstract, this aspect must be clarified.

7.         You have adsorption studies and composite materials characterization studies, thus 3.2.5. Adsorption mechanism must be added in the first part.

8.         Selectivity toward chromium ions using various competitive metals should be added.

9.         Recyclability should be included to explain materials sustainability presented.

10.       Update the bibliography up to the present.

Author Response

Thank you for your thoughtful and constructive review. Due to the large number of answers, I will send them as attachments.

Round 2

Reviewer 1 Report

Comments and Suggestions for Authors

Comments regarding the manuscript entitled "Adsorption of chromium (III) and chromium (VI) ions from aqueous solution using chitosan-clay composite material" (Ref.: polymers-2972553).

The authors considered most of my comments and submitted an improved version of their manuscript, so the revised manuscript may be considered to be accepted for publication. Before publication, please revise the following:

1) There are several typographical errors in the revised manuscript; for example: line 40, leatner; line 176, liner; line 347, C (III); etc.

2) Line 171: The separation factor (KL) units are wrong. 

3) Line 172 and Table 6: The Freundlich constant (KF) units are wrong. Solve KF from the Freundlich model and do the dimensional analysis to obtain the KF units. 

4) The sentence in lines 321-325 is repeated in lines 328-332.

5) Line 423 and Table 6: Is R2 equal to 0.995 or 0.998?

Comments on the Quality of English Language

In general the quality of English is well.

Author Response

We sincerely thank the reviewer for the comments that helped improve our manuscript quality and for your kind advice. 

Following the reviewers' comments and advice, the following has been reviewed.

Further details are attached.

Reviewer 2 Report

Comments and Suggestions for Authors

I read the answers of the authors and they mostly addressed my answers. In my opinion, the article can be accepted for publication.

Author Response

We sincerely thank the reviewer for the comments that helped improve our manuscript quality and for your kind advice.